# Alpha-1 adrenergic receptor antagonists to prevent hyperinflammation and death from lower respiratory tract infection

Allison Koenecke[1†], Michael Powell[2†], Ruoxuan Xiong[3], Zhu Shen[4], Nicole Fischer[5], Sakibul Huq[6], Adham M Khalafallah[6], Marco Trevisan[7], Pär Sparen[7], Juan J Carrero[7], Akihiko Nishimura[8], Brian Caffo[8], Elizabeth A Stuart[8], Renyuan Bai[6], Verena Staedtke[6], David L Thomas[5], Nickolas Papadopoulos[9], Ken W Kinzler[9], Bert Vogelstein[9], Shibin Zhou[9], Chetan Bettegowda[5,9], Maximilian F Konig[10]*, Brett D Mensh[11]*, Joshua T Vogelstein[2,8*†], Susan Athey[12]*

[1]Institute for Computational & Mathematical Engineering, Stanford University, Stanford, United States; [2]Department of Biomedical Engineering, Institute for Computational Medicine, The Johns Hopkins University, Baltimore, United States; [3]Management Science & Engineering, Stanford University, Stanford, United States; [4]Department of Statistics, Stanford University, Stanford, United States; [5]The Johns Hopkins University School of Medicine, Baltimore, United States; [6]Department of Neurosurgery and Neurology, The Johns Hopkins University School of Medicine, Baltimore, United States; [7]Department of Medical Epidemiology and Biostatistics, Karolinska Institutet, Sweden, Solna, Sweden; [8]Department of Biostatistics, Johns Hopkins Bloomberg School of Public Health at Johns Hopkins University, Baltimore, United States; [9]Ludwig Center, Lustgarten Laboratory, and the Howard Hughes Medical Institute at The Johns Hopkins Kimmel Cancer Center, Baltimore, United States; [10]Division of Rheumatology, Department of Medicine, The Johns Hopkins University School of Medicine, Baltimore, United States; [11]Janelia Research Campus, Howard Hughes Medical Institute, Ashburn, United States; [12]Stanford Graduate School of Business, Stanford University, Stanford, United States

*For correspondence:
konig@jhmi.edu (MFK);
bmensh@gmail.com (BDM);
jovo@jhu.edu (JTV);
athey@stanford.edu (SA)

†These authors contributed equally to this work

## Abstract:

In severe viral pneumonia, including Coronavirus disease 2019 (COVID-19), the viral replication phase is often followed by hyperinflammation, which can lead to acute respiratory distress syndrome, multi-organ failure, and death. We previously demonstrated that alpha-1 adrenergic receptor ($\alpha_1$-AR) antagonists can prevent hyperinflammation and death in mice. Here, we conducted retrospective analyses in two cohorts of patients with acute respiratory distress (ARD, n = 18,547) and three cohorts with pneumonia (n = 400,907). Federated across two ARD cohorts, we find that patients exposed to $\alpha_1$-AR antagonists, as compared to unexposed patients, had a 34% relative risk reduction for mechanical ventilation and death (OR = 0.70, p = 0.021). We replicated these methods on three pneumonia cohorts, all with similar effects on both outcomes. All results were robust to sensitivity analyses. These results highlight the urgent need for prospective trials testing whether prophylactic use of $\alpha_1$-AR antagonists ameliorates lower respiratory tract infection-associated hyperinflammation and death, as observed in COVID-19.

**Figure 1.** Model of clinical progression of respiratory dysfunction from local infection to hyperinflammation. The timing and relation of hyperinflammation to specific organ manifestations of severe acute respiratory distress syndrome (ARDS) are areas of uncertainty and investigation.

## Introduction

Each year, approximately 300 million people develop pneumonia (*GBD 2016 Lower Respiratory Infections Collaborators, 2018*), which usually results in appropriate, self-limiting immune responses and clearance of the bacterial or viral pathogen. In some cases (*Figure 1*), the pathogen overwhelms host defenses, causing massive lung damage and compromising other organs. In other patients, however, pathological immune activation ('hyperinflammation') occurs in the lungs and systemically (*Hay et al., 2017*), resulting in immune-mediated end-organ damage that can compromise gas exchange. Dysregulated immune responses may lead to acute respiratory distress syndrome (ARDS), need for mechanical ventilation, and failure of other organ systems, contributing to the global pneumonia death toll of 3 million per year (*The top 10 causes of death, 2021*). The clinical picture is similar in Coronavirus disease 2019 (COVID-19) caused by SARS-CoV-2; hyperinflammation compromises organ function in the lungs and systemically, causing high morbidity and mortality (*Zhou et al., 2020*; *Ruan et al., 2020*; *Qin et al., 2020*; *Huang et al., 2020*).

Disease-modifying strategies for COVID-19 include targeting the virus and treating (or ideally preventing) secondary hyperinflammation with immunomodulatory or immunosuppressive drugs. Here, we propose an approach using the latter strategy of hyperinflammation prevention (*Flierl et al., 2007*; *Shaked et al., 2015*). Immune cells communicate with each other by secreting peptides called cytokines and chemokines, which initially amplify the response and later restore homeostasis after the threat has receded. However, in hyperinflammation, production of cytokines is dysregulated, forming a 'cytokine storm' that can damage healthy tissue and overwhelm the host.

COVID-19-associated hyperinflammation is characterized by profound elevation of many pro-inflammatory cytokines (*Ruan et al., 2020*; *Huang et al., 2020*; *Mehta et al., 2020*; *McGonagle et al., 2020*; *Pedersen and Ho, 2020*; *Chen et al., 2020*). Several immunosuppressive treatment approaches are being studied or used in clinical practice to ameliorate hyperinflammation-associated morbidity in patients who have already developed severe complications of COVID-19, including blocking specific cytokine signaling axes (e.g., IL-6, IL-1, or TNF-alpha) and broader immunosuppressive approaches (e.g. dexamethasone or baricitinib) (*Investigators et al., 2021*; *RECOVERY Collaborative Group, 2021*; *Huet et al., 2020*; *Rosas et al., 2021*; *Salama et al., 2021*; *Stone et al., 2020*). Dexamethasone is one of few drugs that have shown a mortality benefit in hospitalized patients with COVID-19 who require oxygen or mechanical ventilation, but glucocorticoids may not be beneficial - and may even be harmful - when given earlier in the disease course (*RECOVERY Collaborative Group, 2021*; *Gianfrancesco et al., 2020*). Conflicting data on tocilizumab and other inhibitors of IL-6 signaling in patients with severe COVID-19 suggests that immunosuppressive strategies may be of limited benefit once end-organ damage has developed (*Investigators et al., 2021*; *Rosas et al., 2021*; *Salama et al., 2021*; *Stone et al., 2020*; *Lescure et al., 2021*; *Della-Torre et al., 2020*), and highlight the importance of identifying drugs that can prevent dysregulated immune responses and immune-mediated damage.

Inhibition of catecholamine signaling has emerged as a promising approach to prevent hyper-inflammation and related mortality. The catecholamine pathway, beyond its role in neurotransmission and endocrine signaling, is involved in immunomodulation of innate and adaptive immune cells. In mice, catecholamine release coincides with hyperinflammation and enhances inflammatory injury by augmenting cytokine production via a self-amplifying process that requires alpha-1 adrenergic receptor ($\alpha_1$-AR) signaling (*Staedtke et al., 2018*). Catecholamine synthesis inhibition reduces cytokine responses and dramatically increases survival after inflammatory stimuli. The $\alpha_1$-AR antagonist

prazosin (at clinically prescribed dosages)—but not beta-adrenergic receptor (β-AR) antagonists—offers similar protection, providing in vivo evidence that this drug class can prevent cytokine storm (*Staedtke et al., 2018*). These preclinical findings provide a rationale for clinical studies that assess whether $\alpha_1$-AR antagonists can prevent hyperinflammation and its sequelae associated with severe infection.

To date, no controlled trials have studied whether $\alpha_1$-AR antagonism improves clinical outcomes in patients with lower respiratory tract infection (pneumonia, acute respiratory distress syndrome, or COVID-19). Replicated retrospective cohort studies offer the highest level of evidence prior to prospective studies (*Burns et al., 2011*). We thus conducted a series of five retrospective cohort studies, spanning different populations, age groups, demographics, and countries.

Our primary research question is whether $\alpha_1$-AR antagonists (e.g., through its known modulation of hyperinflammation) can mitigate disease and prevent mortality. We operationalized this research question by testing the statistical hypothesis that patients exposed to $\alpha_1$-AR antagonists, as compared to unexposed patients, have a reduced risk of adverse outcomes in lower respiratory tract infection. We considered two outcomes (mechanical ventilation, and mechanical ventilation followed by death) and three exposures (any $\alpha_1$-AR antagonist, tamsulosin specifically, and doxazosin specifically). Tamsulosin is the most commonly used $\alpha_1$-AR antagonist in the United States and demonstrates a 'uroselective' binding pattern (predominantly inhibits $\alpha_{1A}$- and $\alpha_{1D}$-AR subtypes). In contrast, doxazosin is a non-selective $\alpha_1$-AR antagonist that demonstrates clinically significant inhibition of all three known $\alpha_1$-AR subtypes, including antagonism on the $\alpha_{1B}$-AR. This antagonism of $\alpha_{1B}$-AR expressed in the peripheral vasculature is thought to mediate the antihypertensive effects of doxazosin and related drugs. Importantly, all three $\alpha_1$-AR subtypes have been implicated in catecholamine signaling on immune cells, and signaling redundancy suggests a theoretical benefit for pan-$\alpha_1$-AR antagonists (e.g., doxazosin or prazosin) in preventing catecholamine signaling and hyperinflammation. Relatively few patients in our samples are prescribed doxazosin, so we are only able to study this drug in our much larger pneumonia cohorts (since there is insufficient statistical power to analyze the drug in ARD cohorts).

## Results

The statistical analysis plan described in Materials and methods was fixed across all cohorts (to the extent possible) to limit researcher degrees of freedom and to emulate a prospective trial (*Dickerman et al., 2019*). We computed the probabilities of outcomes, relative risk reductions (RRR), odds ratios (OR), confidence intervals (CI), and p-values (p) using an unadjusted model as well as adjusted and matched modeling approaches that account for demographic and health-related confounders. We focus our discussion of reported results on the adjusted model comparing patients exposed to $\alpha_1$-AR antagonists to unexposed patients. The adjusted model is an inverse propensity-weighted regression on a reduced sample satisfying propensity overlap; see Materials and methods for details.

### Participants

We studied two cohorts of patients who were diagnostically coded with acute respiratory distress (ARD, a surrogate precursor state to ARDS) from two de-identified databases: the IBM MarketScan Research Database (which we refer to as MarketScan) and Optum's Clinformatics Data Mart Database (OptumInsight, Eden Prairie, MN), a commercial and Medicare Advantage claims database (which we refer to as Optum). We further studied three cohorts of patients with pneumonia from the MarketScan, Optum, and Swedish National Patient Register (*Patientregistret, 2021*) databases. ICD codes were used to identify the first instance of inpatient admission for each patient in each cohort (see Materials and methods for details). Our main analysis employed federated analyses on the ARD cohort using pooled MarketScan and Optum results.

We limit the study to older men because of the widespread use in the United States of $\alpha_1$-AR antagonists as a treatment for benign prostatic hyperplasia (BPH), a diagnosis clinically unrelated to the respiratory system or immune disorders. Focusing on men over the age of 45 facilitated examining a patient population in which a large portion of the exposed group faced similar risks of poor outcomes from respiratory conditions as the unexposed group, thus mitigating confounding by indication.

We allowed a maximum age of 85 years to reflect the ongoing clinical trials investigating these interventions (Prazosin to Prevent COVID-19 (PREVENT-COVID Trial); https://clinicaltrials.gov/ct2/show/NCT04365257) . *Figure 2* shows the CONSORT flow diagram for selecting patients from the four claims datasets (with slight modifications in the Swedish National Patient Register analysis to reflect different practices in that population); see Materials and methods for details. Focusing this study exclusively on older men limits the study's internal validity to older men, and it would take additional assumptions to justify external validity claims including excluded demographic groups (e.g., women and younger patients). We nevertheless note that there is an important literature on demographic fairness with regard to clinical studies (*Holdcroft, 2007*; *McMurray, 1991*).

## Cohort-specific results

We conducted the same statistical analysis in each of the five cohorts. In each cohort, we measured incidence and odds ratios (OR) for patients exposed to any $\alpha_1$-AR antagonist, or tamsulosin specifically, as compared to unexposed patients, for each outcome. In the two largest pneumonia cohorts, we additionally consider doxazosin exposure. Our main analysis, described in the following section, involves pooling the results from individual cohorts. For the sake of completion, *Figure 3—figure supplements 1–5* show results for each of the five individual cohorts. We generally found a positive RRR given any combination of exposure and outcome, and found that the adjusted model yielded ORs consistently less than 1. In all cohorts, the OR point estimates were robust to model changes, including other doubly robust approaches (*Funk et al., 2011*) such as causal forests (*Wager and Athey, 2018*), each of which yielded similar results to those presented here.

## Federated results

We employed federated causal methods to enable combining patient-level results while maintaining patient privacy and data usage agreement constraints; see Materials and methods for details. We pooled results across MarketScan and Optum for each of the ARD and pneumonia cohorts (the Swedish dataset used a different outcome because ventilation was not reliably coded, so we did not pool these results). For ARD patients (n = 18,547) exposed to any $\alpha_1$-AR antagonist, as compared to unexposed ARD patients, we found for ventilation and death: RRR = 34%, OR = 0.70, 95% CI (0.49–0.99), p = 0.021; for pneumonia patients (n = 338,674) we found: RRR = 8%, OR = 0.86, 95% CI (0.82–0.91), p < 0.001 (*Figure 3*). The treatment effect was similar across the outcome of ventilation only, exposure to tamsulosin only, and other analysis methods (unadjusted and matched), with ORs consistently less than 1. Doxazosin, a non-selective $\alpha_1$-AR antagonist hypothesized to have a greater efficacy than other $\alpha_1$-AR antagonists, demonstrated a twofold stronger effect than tamsulosin, which blocks fewer $\alpha_1$-adrenoreceptors.

## Discussion

The results of this retrospective clinical study extend preclinical findings to support the hypothesis that $\alpha_1$-AR antagonists may reduce morbidity and mortality in patients at risk of hyperinflammation (*Konig et al., 2020*). A challenge resolved by our retrospective analysis is that patients exposed to $\alpha_1$-AR antagonists may differ from unexposed patients in ways that might also affect their outcomes from respiratory diseases. Individuals are often prescribed $\alpha_1$-AR antagonists for chronic diseases, and we consider only patients who used the medications for at least 6 months in the year prior (i.e., for a medical possession ratio of at least 50%) to the index hospital admission for pneumonia or ARDS where we measure patient outcomes. This makes it less likely that the unmeasured severity of respiratory illness at time of admission for our patient population differs substantially between exposed and unexposed groups; however, we cannot rule out such differences. We refer to confounders as factors that vary between patients exposed and unexposed to $\alpha_1$-AR antagonists and also relate to patient outcomes. One important confounder is age, which in turn is related to other health conditions. For this reason, our research design (outlined in more detail in the Materials and methods section) includes several approaches to adjusting for observed demographic and health-related confounders.

Given that older men were generally using $\alpha_1$-AR antagonists for reasons unrelated to ARDS, it was feasible to balance the exposed and unexposed groups on a large set of prognostically important covariates, reducing concerns that differences in outcomes might be due to confounding.

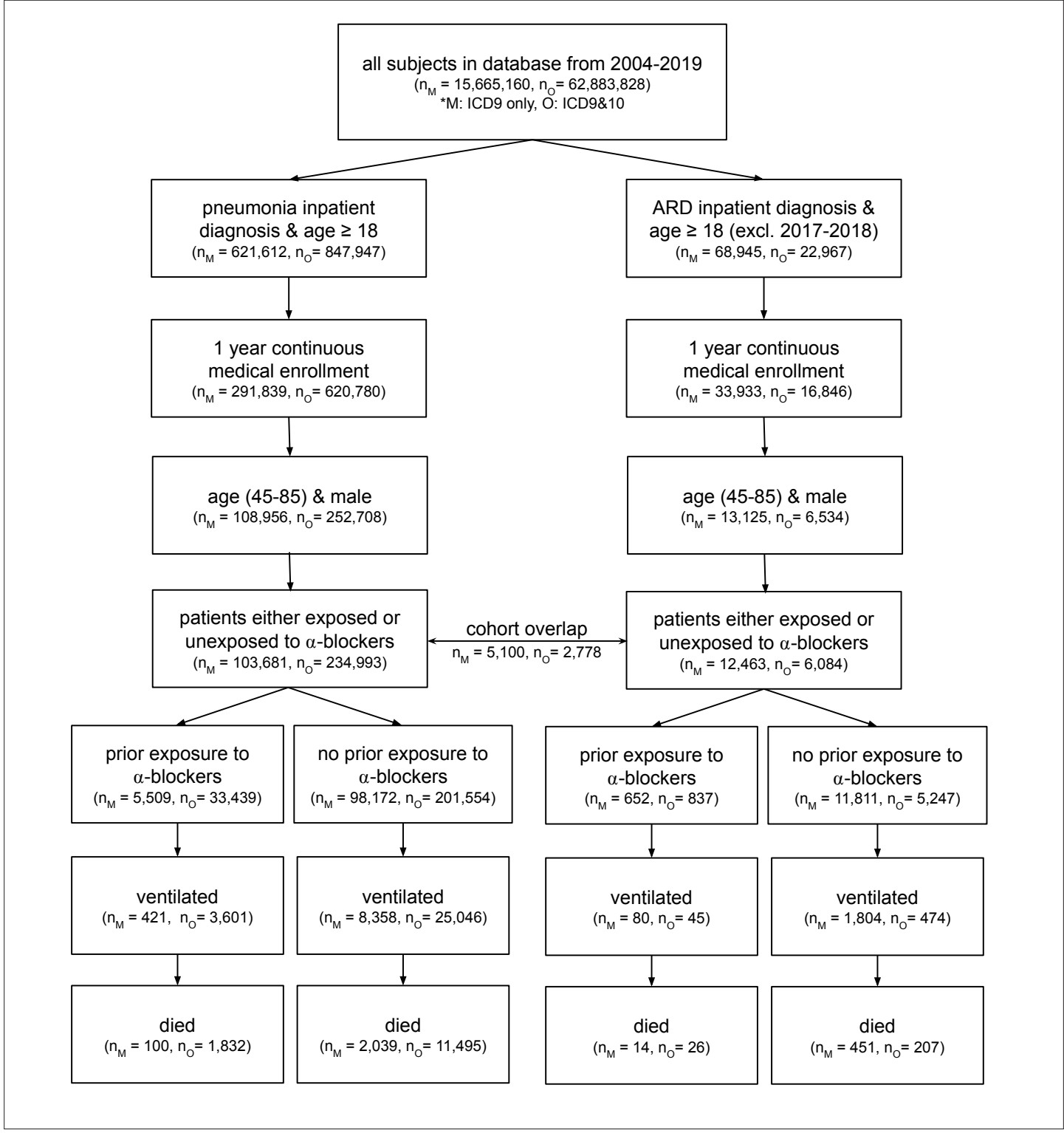

**Figure 2.** CONSORT flow diagram for four claims datasets where M represents MarketScan and O represents Optum; ARD represents acute respiratory distress. Note that patients are considered exposed to $\alpha_1$-AR antagonists if they have a medication possession ratio ≥50 % in the prior year, and are considered unexposed if they have not taken any amount of $\alpha_1$-AR antagonists in the prior year. ARD inpatient visits are not considered between 2017–2018 as ARD ICD-9 codes were being phased out while ICD-10 codes for ARD were not yet commonly used. Within a single dataset (MarketScan or Optum), there exists some patient overlap for the two cohort diagnoses (pneumonia and ARD): 5,100 patients in MarketScan and 2,778 in Optum. This diagram only presents four of the five cohorts studied; the fifth cohort (the Swedish National Patient Register) uses a different set of inclusion/ exclusion criteria (see section Sweden National Patient Register for criteria description, and *Figure 3—figure supplement 1* for information on dataset characteristics).

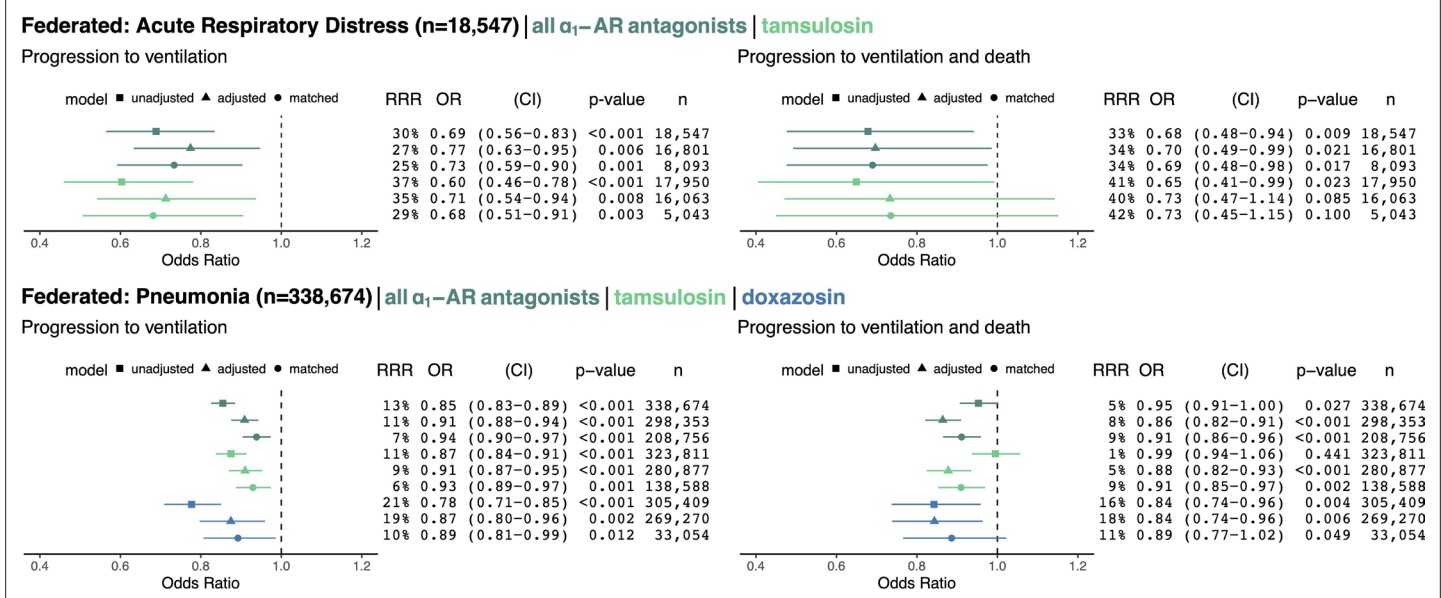

**Figure 3.** Cohorts across datasets (MarketScan and Optum) associated with the same disease (ARD in top row, pneumonia in bottom row) were pooled using federated causal learning techniques described in Materials and methods. In each quadrant, we show: (left) plotted odds ratios (OR) with confidence intervals (CI), and (right) values for relative risk reductions (RRR), OR, CI, p-values (**p**), and sample sizes (**n**) for unadjusted, adjusted, and matched models, including any $\alpha_1$-AR antagonists or specifically tamsulosin or doxazosin. We only study exposure to doxazosin in the pneumonia cohorts since there is insufficient statistical power to analyze the drug in ARD cohorts. Results are shown for outcomes of mechanical ventilation (left column) and mechanical ventilation leading to death (right column). In general, $\alpha_1$-AR antagonists were associated with reducing risk of adverse events across exposures, outcomes, and modeling approaches. Each federated analysis yielded an OR point estimate below 1.

The online version of this article includes the following figure supplement(s) for figure 3:

**Figure supplement 1.** Patients from the Swedish National Patient Register with pneumonia.

**Figure supplement 2.** Patients from MarketScan Research Database with acute respiratory distress.

**Figure supplement 3.** Patients from Optum with acute respiratory distress.

**Figure supplement 4.** Patients from MarketScan Research Database with pneumonia.

**Figure supplement 5.** Patients from Optum with pneumonia.

Nonetheless, there is an urgent need for randomized prospective trials to further test this hypothesis in patients with COVID-19 and other lower respiratory tract infections. In such trials, early administration of $\alpha_1$-AR antagonists prior to development of severe symptoms is preferred because the goal is to prevent, rather than treat, hyperinflammation-related damage. While this study could only examine medications received outside the inpatient environment due to dataset limitations, future work should attempt to address whether continued use of $\alpha_1$-AR antagonists by hospitalized patients is important for preventing severe symptoms. Additionally, future work should consider whether the patients' respiratory illness is community- or hospital-acquired (**Whittle et al., 1997**).

$\alpha_1$-AR antagonists with various receptor subtype characteristics have been used to treat millions of patients with BPH, hypertension, and other disorders. This history supports their safety profile (**Yasukawa et al., 2001**), although caution is warranted in using any medication for the first time in a new disease such as COVID-19. Given the poorly understood relationship between COVID-19 severity and hypertension (**Konig et al., 2020**), it is important to note that nonselective ($\alpha_{1A}=\alpha_{1B}=\alpha_{1D}$) $\alpha_1$-ARs (e.g., prazosin and doxazosin) are often primarily used to reduce blood pressure, whereas subtype-selective ($\alpha_{1A}=\alpha_{1D}>\alpha_{1B}$) $\alpha_1$-ARs (e.g. tamsulosin) have fewer hemodynamic effects. However, given the known redundancy of catecholamine signaling (**Grisanti et al., 2011**), we expect $\alpha_1$-AR antagonists that inhibit all three $\alpha_1$-AR subtypes (e.g. prazosin and doxazosin) to be more efficacious in the prevention of hyperinflammation.

$\alpha_1$-AR antagonists are inexpensive and administered orally, enabling widespread use if prospective trials support their efficacy and safety. Beyond COVID-19 and other lower respiratory tract infections,

$\alpha_1$-AR antagonists may also reduce hyperinflammation and their sequelae in adoptive cell therapy ('cytokine release syndrome') and autoimmune rheumatic disease.

## Materials and methods

### Study definitions

Patients were considered exposed if they were users of $\alpha_1$-AR antagonists (doxazosin, alfuzosin, prazosin, silodosin, terazosin, or tamsulosin). $\alpha_1$-AR antagonist usage is defined by having a medication possession ratio ≥50 % (i.e., a minimum prescribed supply covering six of the last 12 months) for a qualifying drug in the year prior to the first inpatient admission date for ARD or pneumonia. In additional analyses, we used the same definition to consider exposed patients who were taking tamsulosin or doxazosin. Patients are considered unexposed if they have never been prescribed the relevant drug class or drug within the year prior to the admission date. We exclude patients having a medical possession ratio greater than 0% and less than 50% of the relevant drug class or drug (equivalently, we exclude patients who do use $\alpha_1$-AR antagonists, but for less than 6 months of the prior year). While all patients from the MarketScan, Optum, and Swedish National Patient Register (*Patientregistret, 2021*) databases were restricted to a 45–85 year age range, the Market-Scan database only includes patients up to age 65 due to Medicare exclusion; we adjust for age in our analyses. The data used in this study do not include $\alpha_1$-AR antagonist usage during the ARD or pneumonia admission.

The first instances of ARD or pneumonia inpatient admissions for each patient were identified using ICD codes. Because the MarketScan database only includes United States patient data through fiscal year 2016, we only used ICD-9 codes to identify diagnoses from this database. The Optum database includes United States patient data from fiscal year 2004 through 2019, allowing for use of both ICD-9 and ICD-10 codes to identify diagnoses. The Swedish data contained only ICD-10 codes. To identify ARD, we used the ICD-9 code 518.82 and the ensuing ICD-10 code of R.0603, noting that the latter code was only available from 2018 onwards; hence, the Optum ARD cohort included only patients identified by their first inpatient admission for an ARD diagnosis occurring either from 2004 to 2015 or from 2018 to 2019. To identify pneumonia, we used the Agency for Healthcare Research and Quality (AHRQ) pneumonia category for both ICD-9 and ICD-10 codes.

We considered the following potential confounders: age, fiscal year, total weeks with inpatient admissions in the prior year, total outpatient visits in the prior year, total days as an inpatient in the prior year, total weeks with inpatient admissions in the prior two months, and comorbidities identified from healthcare encounters in the prior year: hypertension, ischemic heart disease, acute myocardial infarction, heart failure, chronic obstructive pulmonary disease, diabetes mellitus, and cancer. All comorbidities were defined according to ICD code sets provided by the Chronic Conditions Data Warehouse (*Condition Categories, 2020*). The numeric confounders relevant to prior inpatient and outpatient visits were log-transformed; all confounders were de-meaned and scaled to unit variance.

The outcomes of interest in this study included mechanical ventilation and death. Mechanical ventilation was tabulated as an outcome for patients receiving a procedure corresponding to one of the following ICD-9 codes (967, 9670, 9671, 9672) or ICD-10 codes (5A1935Z, 5A1945Z, 5A1955Z). When tabulating death outcomes, we note a difference in death encoding among databases. From the MarketScan database, we identified death outcomes by in-hospital deaths noted in claims records. From the Optum database, in-hospital death data were unavailable, so we instead identified death outcomes by whether a patient's month of death (if observed) was within 1 month of the relevant ARD or pneumonia inpatient visit month. For both databases, our analyses studied the outcome of death (as defined above) occurring along with mechanical ventilation during inpatient stay. The Swedish National Patient Register's in-hospital deaths were tabulated in a similar manner to the MarketScan database. We focused on death associated with respiratory failure to avoid counting deaths due to other diseases (many of the patients were hospitalized for other reasons), but note that our findings are robust to outcomes defined instead by all-cause mortality, with OR <1. This analysis mirrored our prospective trial design (Prazosin to Prevent COVID-19 (PREVENT-COVID Trial); https://clinicaltrials.gov/ct2/show/NCT04365257).

## Unadjusted analysis

In our baseline unadjusted analysis, we used Fisher's exact test to compare the probability of outcome occurrence with $\alpha_1$-AR antagonist exposure versus without. Fisher's exact test and the associated non-central hypergeometric distribution under the alternative provided the odds ratios, confidence intervals, and p-values (one-sided) reported for individual cohorts. We additionally computed the relative risk reduction (RRR).

## Adjusted analysis

We nonparametrically estimated (using generalized random forests *Athey et al., 2019*) propensity scores and restricted analyses to those individuals with probabilities of being exposed (and of receiving the unexposed condition) between the maximum of the 1st percentiles and the minimum of the 99th percentiles of propensity scores across exposed and unexposed groups (*Stürmer et al., 2010*). This (propensity score trimming) step prevents the comparison of individuals with an extremely low estimated chance of receiving one of the three exposures. We refer to this as our reduced sample. We note that our results are robust to different definitions of propensity score trimming, such as restricting to propensity scores between 0.01 and 0.99.

On the reduced sample, we then apply estimation techniques designed for causal inference to compare the probability of outcome occurrence with $\alpha_1$-AR antagonist exposure versus without. Specifically, we invoke inverse propensity-weighting, which reweights the unexposed population such that on average, the reweighted population more closely resembles the exposed population in terms of observed characteristics. Doing so yields better balancing of covariates, as seen in the pre- and post-inverse-propensity-weighting comparison in *Figure 3—figure supplement 2* (ii)–5(ii). To obtain an odds ratio, we fit an inverse propensity-weighted logistic regression model. This approach is 'doubly robust' in that it yields valid estimates of causal effects if we have (1) observed all relevant confounders and (2) either one of the logistic regression model or the propensity score model is correctly specified. To estimate confidence intervals we used a Wald-type estimator. To obtain one-sided p-values we invoked classical asymptotic theory.

## Matched analysis

An alternative approach comparing patients exposed to $\alpha_1$-AR antagonists with unexposed patients who are as similar as possible—and thus mimicking a randomized controlled trial—involves statistical matching, where each exposed patient is compared to a set of 'nearest neighbors' measured in terms of characteristics. This approach avoids making specific functional form assumptions about the relationship between characteristics and outcomes (*Stuart, 2010*). Specifically, on the reduced sample, we performed a 5:1 matching analysis (*Stuart, 2010*), assigning five unexposed patients to each exposed patient (i.e., patients exposed to any $\alpha_1$-AR antagonist or specifically tamsulosin or doxazosin) and then comparing the outcome of the exposed patient to the average outcome for matched unexposed patients. We required an exact match on patient age and then used a greedy, nearest neighbor approach based on Mahalanobis distance with a caliper of 0.2 on the remaining covariates (*Ho et al., 2011*). We then used the Cochran-Mantel-Haenszel (CMH) test on this matched sample to provide the odds ratios, 95% confidence intervals, and p-values associated with the hypothesis of no effect.

To elaborate, the CMH test can be thought of as a generalization of the chi-square test for association (*Mantel and Haenszel, 1959*). Fisher's exact test considers single 2 × 2 contingency tables containing the counts for two dichotomous variables, specifically an outcome indicator and an exposure indicator (for the setting in this paper). In contrast, the CMH test applies to an array of $k$ contingency tables such that the 2 × 2 × $k$ array of tables represents a stratification of the data into $k$ groups or matched pairs that can be considered comparable (typically groups are defined by the realization of some set of categorical variables) aside from their outcome and exposure values. For each single-source (e.g., MarketScan or Optum) matched model we compiled 2 × 2 contingency tables for exposure and outcome values for the matched pairs obtained exclusively from that database. We then conducted a CMH test on each array of 2 × 2 contingency tables separately. Let $k_M$ represent the number of such tables in the MarketScan data set and $k_O$ be the number of such tables in the Optum data set. In both cases $k$ corresponds to the number of exposed observations for which suitable matches could be identified.

## Federated analysis

Pooling the unadjusted models leveraged the CMH test by considering all observations from each data set to be 'matched' with other observations from the same data set according to one categorical variable: the source database. Thus, each database contributed a single 2 × 2 contingency table to form the 2 × 2 × 2 array evaluated by the CMH test.

To pool the MarketScan and Optum adjusted models, we calculated the pooled coefficient and variance of the exposure using inverse variance weighting (**Hartung et al., 2011**). Let $\hat{\beta}_i$ and $Var(\hat{\beta}_i)$ be the estimated coefficient and variance of the exposure on dataset $i$. The pooled coefficient of the exposure is

$$\hat{\beta}_{pooled} = \frac{\sum_i \hat{\beta}_i / Var(\hat{\beta}_i)}{\sum_i 1 / Var(\hat{\beta}_i)},$$

and the pooled variance of the exposure coefficient is

$$Var(\hat{\beta}_{pooled}) = \frac{1}{\sum_i 1 / Var(\hat{\beta}_i)}.$$

To pool the MarketScan and Optum matched models, we concatenated the 2 × 2 contingency tables for the matched pairs from MarketScan with the 2 × 2 contingency tables for the matched pairs from Optum. In this manner, we effectively pooled the two exposed populations and their matched unexposed populations without ever combining the raw data in any way. Of note, combining raw data was prohibited by both data use agreements. Matches for exposed patients came exclusively from each exposed patient's source data set. The CMH test effectively processed a 2 × 2 × ($k_M$ + $k_O$) array of contingency tables.

Finally, we pooled the relative risk reductions (RRR) estimated from the two data sets using inverse variance weighting of the respective RRRs contributed from separate MarketScan and Optum analyses. First let relative risk reduction (RRR) be defined as

$$\widehat{RRR} = \frac{\hat{p}_y - \hat{p}_x}{\hat{p}_y} = 1 - \frac{\hat{p}_x}{\hat{p}_y},$$

where $X$ corresponds to the exposed group and $Y$ corresponds to the unexposed group. Next, the variance of the RRR estimate can be derived as the variance of a ratio of two binomial proportions:

$$Var(\widehat{RRR}) = Var\left(\frac{\hat{p}_x}{\hat{p}_y}\right) = \frac{1}{n_x} \frac{\hat{p}_x(1-\hat{p}_x)}{\hat{p}_y^2} + \frac{1}{n_y} \frac{\hat{p}_x^2 \cdot \hat{p}_y(1-\hat{p}_y)}{\hat{p}_y^4}.$$

Finally, multiple RRR estimates are pooled using inverse variance weighting:

$$\widehat{RRR}_{pooled} = \frac{\sum_i \widehat{RRR}_i / Var(\widehat{RRR}_i)}{\sum_i 1 / Var(\widehat{RRR}_i)}.$$

## Sensitivity analysis

To assess the robustness of our results to alternative approaches to estimating causal effects under the unconfoundedness assumption (**Steegen et al., 2016**), we explored methods including inverse propensity-weighted (IPW) averaging of outcomes as well as the (doubly robust) augmented inverse propensity-weighted (AIPW) estimator, where we used alternatives such as logistic regression and non-parametric causal forests (**Wager and Athey, 2018**). To assess invariance of our results to definitional choices, we adjusted the definition of exposure to $\alpha_1$-AR antagonists (e.g., requiring regular use of $\alpha_1$-AR antagonists within the prior 3 months rather than 12 months, or excluding $\alpha_1$-AR antagonist users who simultaneously take one of the 12 most common drugs appearing in the cohorts studied) as well as the definitions of certain confounders (e.g., combining three cardiovascular confounders into one indicator, including different metrics of prior inpatient or outpatient stays, including comorbidity severity indices, or including additional comorbidities such as HIV infection). The results using these alternate methods and definitions were consistent with those presented in the results section.

Further, we sought to observe the time course of health decline in the exposed and unexposed groups (quantified as the number of inpatient visits in the months prior to a patient's target admission

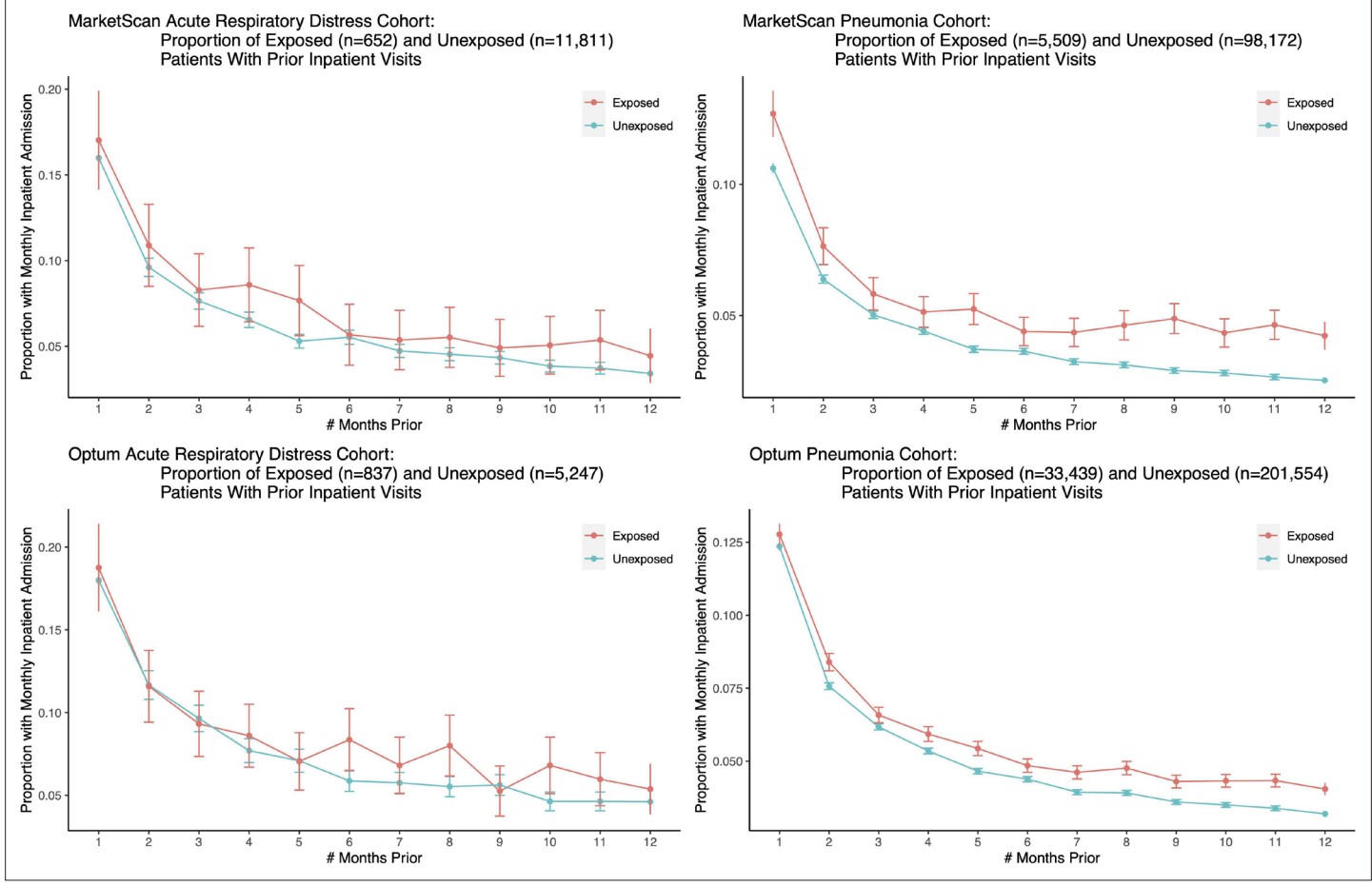

**Figure 4.** We plot the proportion of exposed and unexposed patients having any inpatient admissions a certain number of months prior to the first ARD or pneumonia admission date, and present corresponding confidence intervals. Both exposed and unexposed groups had similar trends of declining health leading up to the target admission date, where health decline is defined as having more frequent inpatient visits.

The online version of this article includes the following figure supplement(s) for figure 4:

**Figure supplement 1.** We plot the average residuals of inpatient visits after controlling for age effects (as well as age squared and age cubed) for exposed and unexposed patients having inpatient admissions a certain number of months prior to the first ARD or pneumonia admission date, and present corresponding confidence intervals.

date). This analysis shows similar temporal trends for the exposed and unexposed groups, indicating that neither group was declining more rapidly than the other; see *Figure 4*.

Lastly, following *Rosenbaum and Rubin, 1983*, we assessed the sensitivity of our estimate of the treatment effect to the presence of an unobserved confounder. We estimated the E-value (*Vander-Weele and Ding, 2017*), which represents 'the minimum strength of association on the risk ratio scale that an unmeasured confounder would need to have with both the treatment and the outcome to fully explain away a specific treatment-outcome association, conditional on the measured covariates' (*VanderWeele and Ding, 2017*). The ORs (and corresponding confidence interval) are defined by the relative odds of occurrence of an outcome (we focus on ventilation and death) given exposure to $\alpha_1$-AR antagonists. We calculated the E-value on both the OR point estimate and confidence interval, each of which has its own interpretation. The minimum odds ratio necessary for an unmeasured confounder (equally associated with both alpha-blocker exposure and the outcome) to move the estimated treatment effect to the null (i.e., OR = 1) is 2.2 for the federated ARD cohort and 1.6 for the federated pneumonia cohort; our findings could not be fully explained by a weaker confounder. The minimum odds ratio necessary for an unmeasured confounder to yield a confidence interval including 1 is an E-value of 1.1 for the federated ARD cohort and 1.4 for the federated pneumonia cohort; we continue to reject the null of no treatment effect in the presence of weaker confounding.

## Sweden National Patient Register

The Swedish National Patient Register differed from the MarketScan and Optum databases in several key ways that motivated slight modifications to the analysis. First, we had access to data from 2006 to 2012, during which time the US used ICD-9 and Sweden used ICD-10. Second, in Sweden, ventilation is not often coded, so we instead only considered the outcome of in-hospital death, which likely underestimates the effect size due to deaths from other causes. Nonetheless, the basic results are preserved in this complementary data set. Results are reported in *Figure 3—figure supplement 1*.

## Acknowledgements

Data for this project were accessed using the Stanford Center for Population Health Sciences Data Core. The PHS Data Core is supported by a National Institutes of Health National Center for Advancing Translational Science Clinical and Translational Science Award (UL1 TR001085) and from Internal Stanford funding. The content is solely the responsibility of the authors and does not necessarily represent the official views of the NIH.We thank Adam Sacarny (Columbia University) for advice on processing and analyzing health care claims data. Dr. Sacarny was not compensated for his assistance. We thank Sandra Aamodt for reviewing and editing the manuscript, and Julia Kuhl and Eric Bridgeford for help generating the figures. We also acknowledge Marc Succhard and Sascha Dublin for helpful discussions. Research, including data analysis, was partially supported by funding from Microsoft Research and Fast Grants, part of the Emergent Ventures Program at The Mercatus Center at George Mason University. Allison Koenecke was supported by the National Science Foundation Graduate Research Fellowship under Grant No. DGE – 1656518. Any opinion, findings, and conclusions or recommendations expressed in this material are those of the authors and do not necessarily reflect the views of the National Science Foundation. Dr. Konig was supported by the National Institute of Arthritis and Musculoskeletal and Skin Diseases of the National Institutes of Health under Award no. T32AR048522. Dr. Bettegowda was supported by the Burroughs Wellcome Career Award for Medical Scientists. Dr. Stuart was supported by the National Institute of Mental Health under Grant R01MH115487. Dr. Staedtke was supported by the National Cancer Institute NCI 5K08CA230179 and 1U01CA247576 and the Sontag Distinguished Scientist Award. Dr. Bai was supported by the National Cancer Institute NCI 1U01CA247576. JJC is supported by the Swedish Research Council (#2019–01059) and the Swedish Heart and Lung Foundation. This work was further supported by The Virginia and DK Ludwig Fund for Cancer Research, The Lustgarten Foundation for Pancreatic Cancer Research, and the BKI Cancer Genetics and Genomics Research Program.

---

## Additional information

### Competing interests

Renyuan Bai, Verena Staedtke: Listed as an inventor on a patent application filed in 2017 by The Johns Hopkins University (JHU) (application number 20200368324) on the use of various drugs to prevent cytokine release syndromes. JHU will not assert patent rights from this filing for treatment related to COVID-19. Nickolas Papadopoulos: Listed as an inventor on a patent application filed in 2017 by The Johns Hopkins University (JHU) (application number 20200368324) on the use of various drugs to prevent cytokine release syndromes. JHU will not assert patent rights from this filing for treatment related to COVID-19. NP is a founder of and holds equity in Thrive Earlier Detection. NP is on the Board of Directors of and a consultant to Thrive Earlier Detection. NP is a founder of, holds equity in, and serves as a consultant to Personal Genome Diagnostics. NP is a consultant to and holds equity in NeoPhore. NP is also an inventor on technologies unrelated or indirectly related to the work described in this article. Licenses to these technologies are or will be associated with equity or royalty payments to the inventors, as well as to JHU. The terms of all these arrangements are being managed by JHU in accordance with its conflict of interest policies.. Ken W Kinzler: Listed as an inventor on a patent application filed in 2017 by The Johns Hopkins University (JHU) (application number 20200368324) on the use of various drugs to prevent cytokine release syndromes. JHU will not assert patent rights from

this filing for treatment related to COVID-19. KWK is a founder of and holds equity in Thrive Earlier Detection. KWK is on the Board of Directors of and a consultant to Thrive Earlier Detection. KWK is a founder of, holds equity in, and serves as a consultant to Personal Genome Diagnostics. KWK is a consultant to Sysmex, Eisai, and CAGE Pharma and holds equity in CAGE Pharma. KWK is a consultant to and holds equity in NeoPhore. KWK is an inventor on technologies unrelated or indirectly related to the work described in this article. Licenses to these technologies are or will be associated with equity or royalty payments to the inventors, as well as to JHU. The terms of all these arrangements are being managed by JHU in accordance with its conflict of interest policies.. Bert Vogelstein: Listed as an inventor on a patent application filed in 2017 by The Johns Hopkins University (JHU) (application number 20200368324) on the use of various drugs to prevent cytokine release syndromes. JHU will not assert patent rights from this filing for treatment related to COVID-19. BV is a founder of and holds equity in Thrive Earlier Detection. BV is a consultant to and holds equity in NeoPhore. BV is a founder of, holds equity in, and serves as a consultant to Personal Genome Diagnostics. BV is a consultant to Sysmex, Eisai, and CAGE Pharma and holds equity in CAGE Pharma. BV is also inventors on technologies unrelated or indirectly related to the work described in this article. Licenses to these technologies are or will be associated with equity or royalty payments to the inventors, as well as to JHU. The terms of all these arrangements are being managed by JHU in accordance with its conflict of interest policies.. Shibin Zhou: Listed as an inventor on a patent application filed in 2017 by The Johns Hopkins University (JHU) (application number 20200368324) on the use of various drugs to prevent cytokine release syndromes. JHU will not assert patent rights from this filing for treatment related to COVID-19. SZ holds equity in Thrive Earlier Detection and has a research agreement with BioMed Valley Discoveries Inc. SZ is a consultant to and holds equity in NeoPhore. SZ is a founder of, holds equity in, and serves as a consultant to Personal Genome Diagnostics.. Chetan Bettegowda: CB is a consultant to Depuy-Synthes and Bionaut Pharmaceuticals. CB is also an inventor on technologies unrelated or indirectly related to the work described in this article. Licenses to these technologies are or will be associated with equity or royalty payments to the inventors, as well as to JHU. The terms of all these arrangements are being managed by JHU in accordance with its conflict of interest policies.. Susan Athey: SA is an advisor and holds an equity stake in two private companies, Prealize (Palo Alto, California, USA) and Dr. Consulta (Brazil). Prealize is a health care analytics company, and Dr. Consulta operates a chain of low-cost medical clinics in Brazil.. The other authors declare that no competing interests exist.

## Funding

| Funder | Grant reference number | Author |
| --- | --- | --- |
| Microsoft Research | | Joshua T Vogelstein |
| George Mason University | Emergent Ventures Program | Joshua T Vogelstein |
| National Science Foundation | DGE – 1656518 | Allison Koenecke |
| National Institute of Arthritis and Musculoskeletal and Skin Diseases | T32AR048522 | Maximilian F Konig |
| Burroughs Wellcome Fund | | Chetan Bettegowda |
| National Institute of Mental Health | R01MH115487 | Elizabeth A Stuart |
| National Cancer Institute | 5K08CA230179 | Verena Staedtke |
| National Cancer Institute | 1U01CA247576 | Verena Staedtke |
| National Cancer Institute | 1U01CA247576 | Renyuan Bai |
| Swedish Research Council | #2019–01059 | Juan J Carrero |
| Swedish Heart-Lung Foundation | | Juan J Carrero |

| Funder | Grant reference number | Author |
|---|---|---|

The funders had no role in study design, data collection and interpretation, or the decision to submit the work for publication.

## Author contributions

Allison Koenecke, Michael Powell, Ruoxuan Xiong, Zhu Shen, Nicole Fischer, Marco Trevisan, Pär Sparen, Juan J Carrero, Akihiko Nishimura, Brian Caffo, Elizabeth A Stuart, Renyuan Bai, Nickolas Papadopoulos, Joshua T Vogelstein, Susan Athey, Conceptualization, Data curation, Formal analysis, Funding acquisition, Investigation, Methodology, Project administration, Resources, Software, Supervision, Validation, Visualization, Writing – original draft, Writing – review and editing; Sakibul Huq, Adham M Khalafallah, Verena Staedtke, Ken W Kinzler, Bert Vogelstein, Shibin Zhou, Chetan Bettegowda, Maximilian F Konig, Brett D Mensh, Conceptualization, Formal analysis, Funding acquisition, Investigation, Methodology, Project administration, Resources, Software, Supervision, Validation, Visualization, Writing – original draft, Writing – review and editing; David L Thomas, Writing – original draft, Writing – review and editing

## Author ORCIDs

Allison Koenecke http://orcid.org/0000-0002-6233-8256
Michael Powell http://orcid.org/0000-0003-2749-3725
Pär Sparen http://orcid.org/0000-0002-5184-8971
Maximilian F Konig http://orcid.org/0000-0001-5045-5255
Joshua T Vogelstein http://orcid.org/0000-0003-2487-6237

## Ethics

No human experiments were performed as this study was retrospective in nature and was conducted using third-party data. Research activity on population health on de-identified data has been judged exempt by the Stanford IRB, precluding consent and other requirements associated with human subjects research. Access to IBM MarketScanx Research Databases and Optum's Clinformatics Data Mart Database was granted through the Stanford Center for Population Health Sciences (PHS); access to the Swedish National Patient Register was approved by the regional ethical review board in Stockholm and the Swedish National Board of Welfare, and informed consent was waived. This research is covered under Stanford PHS protocol 40974.

## Decision letter and Author response

Decision letter https://doi.org/10.7554/eLife.61700.sa1
Author response https://doi.org/10.7554/eLife.61700.sa2

# Additional files

## Supplementary files

- Source code 1. R Markdown code to run retrospective analysis and generate figures.
- Source code 2. R script to generate cohorts and perform propensity matching.
- Transparent reporting form

## Data availability

Data for these analyses were made available to the authors through third-party license from Optum and Truven (2 commercial data providers in the United States), and Socialstyrelsen in Sweden. As such, the authors cannot make these data publicly available due to data use agreement. Other researchers can access these data by purchasing a license through Optum and Truven, and placing a data order through the Swedish National Patient Register. Inclusion criteria specified in the 'Materials and methods' section (specifically, the 'Study Definition' subsection) would allow other researchers to identify the same cohort of patients we used for these analyses. Interested individuals may see https://www.optum.com/solutions/prod-nav/product-data.html for more information on accessing Optum data, https://marketscan.truvenhealth.com/marketscanportal/ for more information on accessing Truven (MarketScan) data, and https://www.socialstyrelsen.se/statistik-och-data/register/alla-register/patientregistret/ for more information on accessing the Swedish National Patient Register.

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
