## [Decision Letter]

**Acceptance summary:**

The manuscript deals with the role of previous treatment with α_1_ adrenergic receptor antagonists in the prevention of mechanical ventilation and death from pneumonia. The provide important results supporting a protective effect of ⍺1-AR antagonists in ARDS and pneumonia.

**Decision letter after peer review:**

Thank you for submitting your article "Alpha-1 adrenergic receptor antagonists to prevent hyperinflammation and death from lower respiratory tract infection" for consideration by *eLife*. Your article has been reviewed by 3 peer reviewers, including Evangelos J Giamarellos-Bourboulis as the Reviewing Editor and Reviewer #3, and the evaluation has been overseen by Jos van der Meer as the Senior Editor. The following individuals involved in review of your submission have agreed to reveal their identity: Jesus Bermejo-Martin (Reviewer #1); Evdoxia Kyriazopoulou (Reviewer #2).

The reviewers have discussed the reviews with one another and the Reviewing Editor has drafted this decision to help you prepare a revised submission.

The manuscript outscores the role of previous treatment with α_1_ adrenergic receptor antagonists in the prevention of mechanical ventilation and death from pneumonia. The manuscript provides interesting results suggesting a protective effect of ⍺_1_-AR antagonists in ARDS and pneumonia I. However, major issues need to be addressed:

1. The type of lower respiratory infection the authors refer to is far too vague; the authors need to focus separately on community-acquired and hospital-acquired infections.

2. We do not agree with matching criteria: there is no doubt that matching is important. However, to be certain of the protective effect of the impact of chronic medication on the outcome of pneumonia, matching should take into consideration comorbidities and severity scores. Although supplementary figures analyse for comorbidities, nothing is done for severity indexes. This is a major limitation.

3. Is HIV infection one of the studied comorbidities?

4. How the authors treat patients who are receiving α_1_ adrenergic receptor antagonists for less than 6 months per year?

5. Figure 1 is very difficult to understand. Not even the abbreviations e.g. PNA are provided.

6. The authors state that "our primary research question is whether ⍺_1_-AR antagonists can mitigate or prevent cytokine-storm syndrome"…Where are the results demonstrating the impact of ⍺_1_-AR antagonists on cytokine levels? The authors demonstrate a statistical protective association between the use of these drugs and the risk of death or mechanical ventilation in patients with ARD or pneumonia. The role of cytokine storm in these syndromes is yet to be elucidated. In example, the authors directly accept the cytokine storm theory in COVID-19. We strongly recommend to revise the title and modulate the work`s objective, since no cytokine data are offered here.

7. How the authors control the potential influence of other treatments? Patients needing of ⍺1-AR antagonists use to be elderly men, and in consequence they often receive multiple chronic treatments.

8. The authors have chosen as outcomes the incidence of mechanical ventilation and the incidence of mechanical ventilation and subsequent death. There is a huge overlap between these two outcomes and moreover, death due to other causes is overseen. We would suggest changing the second outcome with all-cause mortality or adding it as third.

9. The authors describe that they have used five different cohorts of patients. These cohorts are not obviously presented in Figure 2. Moreover, the baseline demographics of patients are nowhere given. Did they differ perhaps in the rate of other comorbidities or severity indexes and these differences confound results? What did they match in their matched analysis?

---

## [Author Response]

The manuscript outscores the role of previous treatment with α_1_ adrenergic receptor antagonists in the prevention of mechanical ventilation and death from pneumonia. The manuscript provides interesting results suggesting a protective effect of α_1_-AR antagonists in ARDS and pneumonia I. However, major issues need to be addressed:1. The type of lower respiratory infection the authors refer to is far too vague; the authors need to focus separately on community-acquired and hospital-acquired infections.

Thank you for raising this important point; we agree that the CAP/HAP separation could be useful if there are differences in outcomes among populations. Unfortunately, making a precise CAP/HAP determination for each patient is not possible in our study using our available claims data; it is also not clear if standard criteria to make this distinction from claims data alone have been defined in the relevant literature. In fact, the literature on identifying pneumonia cases from medical claims data focuses somewhat exclusively on identifying community-acquired or all-cause pneumonia rather than distinguishing CAP/HAP cases [1, 2, 3, 4]. Upon reviewing the relevant CAP-identifying literature, we found that identifying pneumonia cases on a large scale requires automated case identification using administrative data, which – when compared to manual chart review in small studies [3, 5] – exhibits modest sensitivity and specificity. The best distinction we can find for approximating if a case is CAP or HAP using claims data requires knowledge of both the timing of imaging and radiological findings related to time of hospitalization; we have visibility of neither the timing nor the findings of imaging studies as neither exists in our available medical claims data (which only contain overall ICD codes useful for identifying all-cause pneumonia). For example, CAP is defined with pneumonia listed only as the first diagnosis code in Whittle 1997, but Guevara 1999 claims pneumonia in the first five diagnosis code positions greatly increases sensitivity without sacrificing specificity (both heuristics specifically aim to identify CAP cases) [3, 5]. An additional layer of complexity relates to the frequently changing clinical definitions of CAP, HAP, and health care-associated pneumonia (HCAP), a construct abandoned in 2016 [6, 7, 8], over the duration of our retrospective study (2004-2019). In summary, our data limitations, the lack of an accepted process for CAP/HAP determination in medical claims data, and significantly changing clinical definitions over the study period prevent breaking up our all-cause pneumonia analysis.

We have included a note on this topic in the Discussion of our manuscript (Section 3):

“Additionally, future work should consider whether the patients’ respiratory illness is community- or hospital-acquired [5] (a distinction we could not make with our data), since the ensuing severity of ARDS or PNA may differ accordingly.”

2. We do not agree with matching criteria: there is no doubt that matching is important. However, to be certain of the protective effect of the impact of chronic medication on the outcome of pneumonia, matching should take into consideration comorbidities and severity scores. Although supplementary figures analyse for comorbidities, nothing is done for severity indexes. This is a major limitation.

We are uncertain whether the concern over missing severity indices is referring to individual comorbidity severity indices or the more common comorbidity summary indices (e.g., Charlson Comorbidity Index or Elixhauser Comorbidity Index), so we will address both in this response.

We performed several robustness checks that did include the calculated Elixhauser index (using the `comorbidityscores` R package) with available data; these model variants did not meaningfully change our results. We chose to not include summary indices (e.g., the Elixhauser or Charlson) in our main results because we only observe health data in the one year prior, and it may not be reasonable to classify individuals with no inpatient or outpatient visits in the prior year as individuals with perfect health per these indices (a frequent occurrence in our data).

We invested significant time considering the different measures of severity indices found in the prior literature, and chose to use a method aligned with OHDSI [9] as described in this paragraph. It is true that we do not address severity at the level of each comorbidity; this is primarily due to lack of visibility on required data used to compute certain comorbidity severity scores (e.g., laboratory results, questionnaires / physician’s examinations, and certain sociodemographic data). Additionally, adding a severity index as an additional variable for each comorbidity could lead to more instability in our models by dramatically increasing the number of covariates (with severity indices either as additional covariates or as additional levels beyond our current binary representation for each comorbidity). The matching and regression methods used for all analyses do, however, involve nonspecific comorbidity severity measures as is standard per OHDSI. As indicated in the Supplementary Figures, we use covariates including: total weeks with inpatient admissions in the prior year, total outpatient visits in the prior year, total prior days as an inpatient in the prior year, and total weeks with prior inpatient stays in the previous two months (corresponding to STAYS12, VISIT12, DAYS12, and STAYS2, respectively). In addition, our matching criteria require an exact match on patient age and a nearest-neighbor match on all remaining covariates, which include comorbidities and the log-transformed values of the severity measures listed above.

We include the following addendum to our manuscript in Section 4.6:

“To assess invariance of our results to definitional choices, we adjusted the definition of exposure to α_1_-AR antagonists (e.g., requiring regular use of α_1_-AR antagonists within the prior 3 months rather than 12 months, or excluding α_1_-AR antagonist users who simultaneously take one of the 12 most common drugs appearing in the cohorts studied) as well as the definitions of certain confounders (e.g., combining three cardiovascular confounders into one indicator, including different metrics of prior inpatient or outpatient stays, **including comorbidity severity indices,** or including additional comorbidities such as human immunodeficiency virus [HIV] infection).”

3. Is HIV infection one of the studied comorbidities?

We have confirmed our results do not meaningfully change when including HIV (ICD codes 042 and B20) as a covariate; see Author response image 1. We did not include HIV infection as a comorbidity in our main analysis since there is extremely rare (<1%) incidence of HIV in our cohorts potentially leading to instability in matching; but, our work is broadly robust to different sets of relevant comorbidities (e.g., using a broader category of cardiovascular comorbidities rather than our chosen selection).

As noted in our response to the second point above, we include the following addendum to our manuscript in Section 4.6:

“To assess invariance of our results to definitional choices, we adjusted the definition of exposure to α_1_-AR antagonists (e.g., requiring regular use of α_1_-AR antagonists within the prior 3 months rather than 12 months, or excluding ⍺_1_-AR antagonist users who simultaneously take one of the 12 most common drugs appearing in the cohorts studied) as well as the definitions of certain confounders (e.g., combining three cardiovascular confounders into one indicator, including different metrics of prior inpatient or outpatient stays, including comorbidity severity indices, or including additional comorbidities such as human immunodeficiency virus [HIV] infection).”

4. How the authors treat patients who are receiving α_1_ adrenergic receptor antagonists for less than 6 months per year?

These few patients are removed from the retrospective analysis entirely, so our treatment effect is based on directly comparing patients known to be long-term users of alpha blockers (i.e., using for more than 6 months of the year) to patients known to have never in the prior year used alpha blockers.

This is clarified in Section 4.1 of the manuscript:

“We exclude patients having a medical possession ratio greater than 0% and less than 50% of the relevant drug class or drug (equivalently, we exclude patients who do use ⍺1-AR antagonists, but for less than six months of the prior year).”

5. Figure 1 is very difficult to understand. Not even the abbreviations e.g. PNA are provided.

Thank you for noting this; we have revised the Figure 1 caption accordingly to define ARDS, and we updated Figure 2 to eliminate the PNA abbreviation and include more explanation in the caption.

We additionally remove the PNA abbreviation from the paper writ large, and use the term ‘pneumonia’ instead.

6. The authors state that "our primary research question is whether α_1_-AR antagonists can mitigate or prevent cytokine-storm syndrome"…Where are the results demonstrating the impact of α_1_-AR antagonists on cytokine levels? The authors demonstrate a statistical protective association between the use of these drugs and the risk of death or mechanical ventilation in patients with ARD or pneumonia. The role of cytokine storm in these syndromes is yet to be elucidated. In example, the authors directly accept the cytokine storm theory in COVID-19. We strongly recommend to revise the title and modulate the work`s objective, since no cytokine data are offered here.

Thank you for this helpful contextualization; we believe that, unfortunately, the reviewers were sent our initial submission rather than our final submission to review. Your comments were resolved in our previous (i.e., ‘final’ submission) to *eLife*. While tracked changes in the manuscript will not reflect these changes, we left comments on the document where the relevant language exists in the Introduction and Discussion.

To clarify, we received similar feedback upon first submitting the article suggesting we reduce references to cytokine storms as we do not directly study cytokine levels nor have the ability to measure them in claims data. Before submitting our final manuscript, we incorporated this feedback by changing the title and introductory and discussion paragraphs to reflect that our hypothesis addresses the role of alpha blockers in lower respiratory tract infections, a hypothesis we connect to our prior cytokine storm study in mice. It appears that the version of our paper that went to reviewers did not reflect these changes (the quoted sentence identified by the reviewer does not exist verbatim in our most recently submitted manuscript).

We also agree with the reviewer that our understanding of immunopathogenesis and the nomenclature of cytokine dysregulation in patients with severe COVID-19 and other respiratory tract infections is evolving. We have used cytokine storm syndrome as an umbrella term to designate hyperinflammation rather than more narrowly used terms such as cytokine release syndrome (CRS) as seen in CAR-T cell therapy, hemophagocytic lymphohistiocytosis (HLH) with perforin pathway loss of function mutations, or macrophage activation syndrome (MAS) in patients with systemic juvenile idiopathic arthritis, which have phenotypic similarities and overlapping but likely not identical immunopathogeneses [10]. To address the reviewer’s concerns, we have changed the nomenclature in the manuscript to hyperinflammation when extrapolating to mechanism.

7. How the authors control the potential influence of other treatments? Patients needing of α_1_-AR antagonists use to be elderly men, and in consequence they often receive multiple chronic treatments.

We control for confounders in our study (i.e., the comorbidities that would necessitate other treatments). We do not control for treatments themselves given the high dimensionality of potential treatments, the relatively small number of non-alpha-blocker treatments being taken consistently and in tandem with alpha-blockers, and the lack of evidence that other treatments would impact respiratory outcomes in a way that would not be accounted for by the relevant comorbidity.

Because, as you mention, the patients needing ⍺_1_-AR antagonists are often elderly men, we do in particular restrict our analyses to men over the age of 45, and adjust for age, age squared, and age cubed in the adjusted analyses (Section 4.3); we require an exact match on age in the matching analyses (Section 4.4).

Additionally, we have run robustness checks where our analyses are conducted for a treatment defined as taking ⍺_1_-AR antagonists and individually excluding each of the most commonly taken drugs among the relevant patients (furosemide, simvastatin, amlodipine, atorvastatin, lisinopril, potassium chloride, carvedilol, omeprazole, allopurinol, gabapentin, levothyroxine sodium, and metoprolol); roughly 70-80% of ⍺_1_-AR antagonist takers are also taking one of these additional drugs, so we exclude users of these drugs one drug at a time in separate analyses. The results for each drug exclusion are consistent with the main findings presented in our paper, wherein treatment involving ⍺_1_-AR antagonists yields odds ratios < 1 for both relevant outcomes (mechanical ventilation and subsequent death).

As noted in our response to the second point above, we include a comment on this in Section 4.6:

“To assess invariance of our results to definitional choices, we adjusted the definition of exposure to α_1_-AR antagonists (e.g., requiring regular use of α_1_-AR antagonists within the prior 3 months rather than 12 months, or **excluding α_1_-AR antagonist users who simultaneously take one of the 12 most common drugs appearing in the cohorts studied) as** well as the definitions of certain confounders (e.g., combining three cardiovascular confounders into one indicator, including different metrics of prior inpatient or outpatient stays, including comorbidity severity indices, or including additional comorbidities such as human immunodeficiency virus [HIV] infection).”

8. The authors have chosen as outcomes the incidence of mechanical ventilation and the incidence of mechanical ventilation and subsequent death. There is a huge overlap between these two outcomes and moreover, death due to other causes is overseen. We would suggest changing the second outcome with all-cause mortality or adding it as third.

The goal of our analysis is to study outcomes related to respiratory illness; the most direct way to study mortality outcomes in this context is to count deaths that coincide with mechanical ventilation. As such, we do not believe all-cause mortality is as relevant a metric to study our particular research question.

That said, we have found that our results are robust to additional outcomes, such as all-cause mortality and the combined outcome of death *or* ventilation (see Author response image 2); the resulting odds ratios are in line with our findings based on the metrics of ventilation, and death *and* ventilation.

We have amended our manuscript in Section 4.1 to include:

“We focused on death associated with respiratory failure to avoid counting deaths due to other diseases (many of the patients were hospitalized for other reasons), but note that our findings are robust to outcomes defined instead by all-cause mortality, with OR < 1.”

9. The authors describe that they have used five different cohorts of patients. These cohorts are not obviously presented in Figure 2. Moreover, the baseline demographics of patients are nowhere given. Did they differ perhaps in the rate of other comorbidities or severity indexes and these differences confound results? What did they match in their matched analysis?

Thank you for raising this concern. Figure 2 only contains statistics for four cohorts – pneumonia patients from the Marketscan and Optum databases, and ARD patients from the Marketscan and Optum databases (corresponding to n_M_, n_O_ under the left branch of the CONSORT diagram, and n_M_, n_O_ under the right branch of the CONSORT diagram, respectively); we had excluded counts on the fifth cohort (Swedish National Patient Register) because a different set of inclusion/exclusion criteria were used (see section 4.7).

We augment the Figure 2 caption as follows:

“This diagram only regards four of the five cohorts studied; the fifth cohort (the Swedish National Patient Register) uses a different set of inclusion/exclusion criteria (see Section 4.7 for criteria description, and Supplementary Figure 3—figure supplement 1 for information on dataset characteristics).”

Baseline demographics of the patients for each cohort were shown in panel (i) of Supplementary Figure 3—figure supplements 2-5; we have augmented Supplementary Figure 3—figure supplement 1 to include demographic distributions for the pneumonia cohort from the Swedish National Patient Register database (which includes all males, aged 45-85). The primary driver of demographic differences among cohorts is that the MarketScan database only includes patients up to age 65, whereas the other cohorts include patients up to age 85 (see description in section 4.1); as such, the MarketScan cohorts tend to have generally healthier patients. However, we resolve this concern by adjusting for confounders such as age and comorbidity. For the Mahalanobis matching analysis (described in point #2 of this response), each cohort only matches within itself.

**References:**

[1] Yu O, Nelson JC, Bounds L, Jackson LA. Classification algorithms to improve the accuracy of identifying patients hospitalized with community-acquired pneumonia using administrative data. Epidemiology and Infection. 2011 Sep;139(9):1296-306.

[2] van de Garde EM, Oosterheert JJ, Bonten M, Kaplan RC, Leufkens HG. International classification of diseases codes showed modest sensitivity for detecting community-acquired pneumonia. Journal of clinical epidemiology. 2007 Aug 1;60(8):834-8.

[3] Guevara RE, Butler JC, Marston BJ, Plouffe JF, File Jr TM, Breiman RF. Accuracy of ICD-9-CM codes in detecting community-acquired pneumococcal pneumonia for incidence and vaccine efficacy studies. American journal of epidemiology. 1999 Feb 1;149(3):282-9.

[4] Skull SA, Andrews RM, Byrnes GB, Campbell DA, Nolan TM, Brown GV, Kelly HA. ICD-10 codes are a valid tool for identification of pneumonia in hospitalized patients aged⩾ 65 years. Epidemiology and Infection. 2008 Feb;136(2):232-40.

[5] Whittle J, Fine MJ, Joyce DZ, Lave JR, Young WW, Hough LJ, Kapoor WN. Community-acquired pneumonia: can it be defined with claims data?. American Journal of Medical Quality. 1997 Dec;12(4):187-93.

[6] American Thoracic Society. Guidelines for the Management of Adults with Hospital-acquired, Ventilator-associated, and Healthcare-associated Pneumonia. American Journal of Respiratory and Critical Care Medicine. 2004 Oct;171:388-416.

[7] Kalil A, Metersky M, Klompas M, Muscedere J, Sweeney D, Palmer L, Napolitano L, O’Grady N, Bartlett J, Carratalà J, El Solh A, Ewig S, Fey P, File T, Restrepo M, Roberts J, Waterer G, Cruse P, Knight S, Brozek J. Management of Adults with Hospital-acquired and Ventilator-associated Pneumonia: 2016 Clinical Practice Guidelines by the Infectious Diseases Society of America and the American Thoracic Society. Clinical Infectious Diseases. 2016 Jul;63(5):61-111.

[8] Kumar S, Yassin A, Bhowmick T, Dixit D. Recommendations From the 2016 Guidelines for the Management of Adults with Hospital-Acquired or Ventilator-Associated Pneumonia. Pharmacy and Therapeutics. 2017 Dec;42(12):767-772.

[9] Reps J and Chen Y. Applying the Lossless Distributed Linear Mixed Model to integrate heterogeneous COVID-19 hospitalization data across the OHDSI Network. OHDSI Studies. 2020 Aug.

[10] Henderson LA, Canna SW, Schulert GS, Volpi S, Lee PY, Kernan KF, Caricchio R, Mahmud S, Hazen MM, Halyabar O, Hoyt KJ. On the alert for cytokine storm: immunopathology in COVID‐19. Arthritis and Rheumatology. 2020 Jul;72(7):1059-63.